# Endoscopic Surveillance of Esophageal Atresia Population according to ESPGHAN-NASPGHAN 2016 Guidelines: Incidence of Eosinophilic Esophagitis and New Histological Findings

Francesca Maestri [1,2,*], Anna Morandi [1], Martina Ichino [1], Giorgio Fava [1], Giacomo Cavallaro [3], Ernesto Leva [1,2] and Francesco Macchini [1]

1 Department of Pediatric Surgery, Fondazione IRCCS Ca' Granda Ospedale Maggiore Policlinico, 20122 Milan, Italy
2 Department of Clinical Sciences and Community Health, Università degli Studi di Milano, 20122 Milan, Italy
3 Neonatal Intensive Care Unit, Fondazione IRCCS Ca' Granda Ospedale Maggiore Policlinico, 20122 Milan, Italy
* Correspondence: maestri.francesca@gmail.com; Tel.: +39-338-3756565

**Abstract:** Follow-up of children born with esophageal atresia (EA) is mandatory due to high incidence of comorbidities. We evaluated endoscopic findings at follow-up of EA patients performed at our Centre according to ESPGHAN-NASPGHAN 2016 guidelines. A retrospective observational study was performed using data from January 2016 to January 2021. We included EA patients (age range: 1–18 years) who were offered a program of endoscopic and histological high gastrointestinal (GI) tract examinations as per ESPGHAN-NASPGHAN 2016 guidelines. Clinical, surgical, auxological, endoscopic, and histological data were reviewed; variables as polyhydramnios, EA type, surgical type, enteral feeding introduction age, growth data, and symptoms were correlated to endoscopic and histological findings. The population included 75 patients (47 males), with mean age of $5 \pm 4$ years. In 40/75 (53.3%) patients, we recorded oral feeding problems, and upper gastrointestinal or respiratory symptoms suspicious of gastroesophageal reflux. Eosinophilic esophagitis (EoE) incidence was 9/75 (12%), significantly higher than in general population ($p < 0.0001$), and 10/75 (13.3%) presented non-specific duodenal mucosal lesions. EoE represents a frequent comorbidity of EA, as previously known. EA is also burdened by high, never-described incidence of non-specific duodenal mucosal lesions. Embedding high GI tract biopsies in EA endoscopic follow-up should be mandatory from pediatric age.

**Keywords:** esophageal atresia; endoscopy; follow-up; eosinophilic esophagitis; duodenal mucosal lesions; transition of care

## 1. Introduction

Esophageal atresia (EA) is a rare congenital anomaly with a survival rate of 99% in full-term neonates with isolated disease [1]. Modern discussion in the scientific community focuses on the long-term follow-up of the EA population. Most EA patients suffer from gastrointestinal (GI) conditions during their lifetime: 22 to 45% of EA patients have gastroesophageal reflux disease (GERD), up to 17% have eosinophilic esophagitis (EoE), and 9 to 41% have physical growth impairment, mainly in the first years of life [2–7]. It is thus well known that patients born with EA require a lifetime follow-up comprising endoscopic examinations from childhood [8].

In 2016, ESPGHAN and NASPGHAN published guidelines for the follow-up of gastrointestinal and nutritional complications in EA patients from the neonatal period throughout childhood, adolescence, and adulthood [3]. Accordingly, our institution implemented

the ongoing follow-up program for EA patients with ESPGHAN-NASPGHAN recommendations in 2016. Moreover, recent reports hypothesize a possible correlation between eosinophilic esophagitis (EoE) and celiac disease (CD) based on common etiopathogenetic aspects [9]. Both are immune-mediated inflammatory diseases caused by an imbalance in the pathway of Th1/Th2 lymphocytic inflammatory response, which occurs in patients with genetic predisposition after exposure to external triggers, mainly food. Therefore, given the high prevalence of EoE in patients with EA and the thriving difficulties often faced by those patients, in order to rule out eosinophilic gastroenteritis and CD, multiple duodenal biopsies are routinely performed along with esophageal biopsies in the endoscopic follow-up at our Centre.

With our study, we evaluated the findings of the endoscopic and histological follow-up of EA patients as recommended by the ESPGHAN-NASPGHAN 2016 guidelines and assessed the results.

## 2. Materials and Methods

We performed a retrospective observational study on patients born with EA (classified as type A, B, C, or D, according to Gross classification [2]) and receiving endoscopic follow-up between June 2016 and June 2021 at our Centre.

EA patients are treated with Proton Pump Inhibitors (PPIs) for the first year of life. Patients start weaning at around six months of life. At one year of life, PPIs are stopped; patients undergo esophageal 24 h pH monitoring and esophagogastroduodenoscopy (EGD) with biopsies to evaluate the further need for PPIs therapy. EGD is routinely repeated within the third year of life, before school-age (around the sixth year of life), before adolescence (around the tenth year of life), and before transitioning to adulthood; endoscopic evaluations are performed more frequently in symptomatic patients who require PPIs treatment or anti-reflux surgery.

Our clinical follow-up was implemented with the ESPGHAN-NASPGHAN recommendations [3] regarding endoscopic and histological surveillance since their publication in 2016, and performed also duodenal biopsies along with esophageal sampling as we consider it to be good clinical practice. An age- and a weight-appropriate endoscope is used, and examinations are carried out after midazolam oral or intravenous administration [10].

We included all the patients older than one year, on an unrestricted diet, who underwent at least one endoscopic examination with esophageal and duodenal biopsies. Children receiving non-oral enteral feeding at the time of endoscopy were excluded.

Data about type of EA, associated malformations and syndromes, presence of polyhydramnios during pregnancy, gestational age at birth, birth weight, timing and type of esophageal surgery, timing of oral/enteral feeding introduction, need and number of esophageal dilations were collected. In addition, we recorded auxological data at the time of endoscopy. Weight, height, and body mass index (BMI) were compared to WHO growth charts [11]. BMI values below the 15th percentile were consistent with poor growth [12]. In order to evaluate growth curves in homogeneous categories, we divided our population into three groups according to age at the endoscopy: 1- to 5-year-old, ≥5- to 10-year-old, and ≥10-year-old patients. Weight percentiles are unavailable for patients over ten years old [12]. Before the endoscopic examination, we recorded the PPI treatment and the presence of symptoms such as dysphagia, globus pharyngeus, regurgitation, frequent vomit or retching, heartburn, dyspepsia, and recurrent respiratory tract infections. We also considered if the patient had a history of allergy. In addition, we recorded macroscopic findings of the endoscopic examination, as well as the associated histological reports. The macroscopic degree of esophageal inflammation due to GERD was measured with the Hetzel-Dent classification [13]; esophageal samples were analyzed for the presence (and degree) of inflammation and eosinophils per High Power Field. Patients who presented allergic symptoms or EoE symptoms underwent full allergic skin tests. The diagnosis of EoE was made according to the updated criteria of the AGREE Conference [7,14]. We diagnosed EoE in patients who had clinical symptoms and presented esophageal samples with

>15 eosinophils/High Power Field (HPF), in the absence of other possible causes. Gastric and duodenal samples were taken to rule out eosinophilic gastroenteritis. Blood eosinophil count was obtained only in patients with samples suggestive for EoE. Gastric samples were reviewed for peptic damage, Helicobacter pylori, and eosinophils/HPF. Duodenal biopsies were reviewed for eosinophils/HPF, and also assigned a Marsh-Oberhuber grade [15]. CD was diagnosed if duodenal samples matched with symptoms, positive celiac antibodies, and HLA-DQ2/DQ8 presence [16]. Isolated duodenal histology classified as grade $\geq 1$ of the Marsh-Oberhuber classification was considered as non-specific duodenal mucosal lesions [17,18].

The primary endpoint evaluated the incidence of endoscopic pathological findings in our EA population. The secondary endpoint assessed patients' clinical status by examining their growth at the endoscopic time, the presence of symptoms, and the need for PPI treatments.

Results are shown as absolute numbers and percentages, with average coupled with standard deviation and median with range. Inferential statistics analyses were performed, using Fisher's exact test for categorical variables and Mann-Whitney U test for continuous variables; *p*-value < 0.05 was considered significant. Microsoft Excel was used for statistical analysis.

## 3. Results

At our Centre, 126 patients with EA are on active follow-up. Seventy-five patients met the inclusion criteria and were included in the study, with a mean follow-up time of $5 \pm 4$ years. The demographic characteristics of the examined sample are shown in Table 1.

**Table 1.** Demographic characteristics of the examined sample: sex, type of EA according to Gross classification [2], presence of coexisting anomalies.

| Demographic Characteristics | *n* (%) or Mean $\pm$ SD |
|:---:|:---:|
| Sex | Male 47 (62.7) |
| Gross type of EA | A 4 (5.3)<br>B 3 (4)<br>C 65 (86.7)<br>D 3 (4) |
| Polyhydramnios | 29 (38.6) |
| Gestational age at birth | $36 \pm 2.5$ weeks |
| Birth weight | $2361 \pm 647.5$ g |
| Associated anomalies * | 36 (48) |

* VACTERL anomalies in 30, CHARGE association in one, duodenal atresia in one, intestinal malrotation in one, macroglossia in one, giant-cell hepatitis in one, Goldenhar's syndrome in one, hypospadias in 3, and Down's syndrome in 3 patients. *n*: number; SD: standard deviation.

The mean age at first surgery was $2 \pm 1.1$ days and consisted of primary repair of EA in 61 (81.3%) patients. The final surgical anastomosis was esophageal in 68 (90.7%), while 7 (9.3%) patients underwent gastric transposition. Enteral feeding was started on average $12 \pm 7.5$ days after the first surgical treatment. In patients receiving primary correction, oral feeding was started on average $13 \pm 7.5$ days after surgery. In comparison, in patients who underwent gastrostomy, enteral feeding was started on average $9 \pm 7.3$ days postoperatively.

Forty-two (56%) patients developed anastomotic strictures requiring a mean of $3.1 \pm 4.4$ esophageal dilations (range: 1–30), with 10 patients requiring more than 3 dilations; one patient was also affected by congenital esophageal stenosis that was treated with dilations.

Table 2 shows the number of patients for each age group, along with the number of observations within the 15th percentile regarding body weight, height, and BMI at the time of endoscopy.

**Table 2.** Distribution of patient age at the time of endoscopy and number of patients of each group within the 15th percentile of weight, height, and BMI.

| Age at Endoscopy | 1- to 5-Year-Old | $\geq$5 and <10-Year-Old | $\geq$10-Year-Old |
|---|---|---|---|
| *n* (%) | 45 (60%) | 22 (29.3%) | 8 (10.7%) |
| <15th percentile | | | |
| Weight | 17/45 (37.7%) | 7/22 (31.8%) | * |
| Height | 15/45 (33.3%) | 7/22 (31.8%) | 3/8 (37.5%) |
| BMI | 7/45 (15.5%) | 5/22 (22.7%) | 4/8 (50%) |

* Reference WHO growth charts are not available for the body weight of children older than 10 years old [12].

At the endoscopic examination, only 4 (0.5%) patients were on PPI treatment, ranging from 13 months to 17 months. Dysphagia and recurrent respiratory tract infections were present in 3 (75%) patients treated with PPI; 37 (52.1%) patients without treatment presented symptoms such as dysphagia, globus pharyngeus, regurgitation, frequent vomit or retching, heartburn, dyspepsia, and recurrent respiratory tract infections. None of the patients presented with a history of allergy to food, pollen, or fur. All patients presenting with EoE symptoms provided negative results at the allergic skin tests.

Macroscopical signs of GERD diseases were present in 27 (36%) patients and were classified as grade 1 of Hetzel-Dent classification in 24 (32%) cases, grade 2 in 2 (2.7%) cases, and grade 3 in one (1.3%) case. Histological esophagitis was present in 54 (72%) cases. There were no reports of esophageal metaplasia or dysplasia. EoE was diagnosed in 9 (12%) patients, indicating an incidence of EoE in EA patients significantly higher than in the general population ($p \leq 0.0001$). None of the patient diagnosed with EoE were on PPI treatment. The incidence of EoE was 23.5% (4/17) in patients born with long gap defect and 8.6% (5/58) in patients born with non-long gap EA ($p = 0.1958$).

No patient received a histological diagnosis of peptic gastritis, duodenitis, or eosinophilic gastroenteritis.

Duodenal mucosal lesions were found in 10 (13.8%) patients: 8 (11.1%) duodenal samples were assigned a Marsh-Oberhuber grade 1, 1 (1.3%) a grade 2, and 1 (1.3%) a grade 3a. Only the patient with duodenal histology grade 3a received a diagnosis of CD due to matching histological samples and serological results, according to current protocols.

The difference in the incidence of histological findings among EA without distal tracheoesophageal (TE) fistula and EA with distal TE fistula are shown in Table 3.

**Table 3.** The difference in the incidence of histological damages among EA without distal TE fistula and EA with distal TE fistula.

| | EA Type A and B | EA Type C and D | *p*-Value |
|---|---|---|---|
| Histological esophagitis | 6/7 (85.7%) | 48/68 (70.5%) | 0.6654 |
| Eosinophilic esophagitis | 3/7 (42.8%) | 6/68 (8.8%) | **0.0335** |
| Duodenal lesions | 2/7 (28.5%) | 8/68 (11.7%) | 0.2331 |

Among the 35 asymptomatic patients at the endoscopic examination, 4 were diagnosed with EoE, and 3 were found to have pathological duodenal samples.

In the cohort of patients with body weight under the 15th percentile, enteral feeding was introduced on average $11 \pm 4.6$ days after the first surgery; within this group, oral feeding was introduced on average $11 \pm 3.2$ days after esophageal primary correction. In the cohort of patients with body weight above the 15th percentile, enteral feeding was introduced on average $12 \pm 9.1$ days after the first surgery; within this group, oral feeding was introduced on average $13 \pm 9.2$ days after esophageal primary anastomosis. The difference in timing of introduction of alimentation was not significant both in enteral ($p = 0.576$) and in oral feeding ($p = 0.653$).

## 4. Discussion

In our cohort of study, data about the type of defect, prenatal and postnatal features, and associated conditions overlap the general characteristics of EA populations [1,2,8,19].

Many factors contribute to malnutrition and poor growth in the EA population. Indeed, during neonatal and early infancy, associated cardiac and renal diseases, surgical complications (i.e., recurrent tracheoesophageal fistula and anastomotic strictures), need for further surgeries, and lengthy hospitalization can be recalled for poor growth [2,20].

In ESPGHAN-NASPGHAN guidelines, early enteral/oral nutrition coupled with intensive neonatal care is recommended to promote physical growth and reduce the risk of long-term malnourishment [3]. However, in our cohort, there was no difference in the timing of enteral feeding between the underweight children and the patients with appropriate weight-for-age values.

In EA patients, EoE is reported as 17% [4,7] and attributed to a possible genetic association and barrier defect in the esophageal mucosa caused by acid reflux or prolonged exposure to PPI therapy [3]. In this cohort of patients, symptoms of EoE are often misdiagnosed as GERD. Thus, before proceeding with anti-reflux surgery, it is mandatory to exclude eosinophilic esophagitis in patients of all ages with EA [3]. Our group's EoE incidence of 12% is consistent with the literature data. Therefore, our results confirm the need to maintain a high clinical suspicion of EoE for patients in follow-up for EA and to always complete the endoscopic examination with multiple esophageal biopsies. In the natural history of the disease, the greatest risk is represented by the development of esophageal stricture, which occurs in about 25% of patients with EoE [21]. In patients born with EA, an inflammation establishing on a malformed and reconstructed esophagus could increase the incidence of complications such as anastomotic stenosis and gastroesophageal reflux, worsen dysphagia, and contribute to the poor growth frequently observed in these patients.

To the best of our knowledge, it is the first time that the high incidence of 42.8% of EoE in patients born with EA type A and B was reported, a value that is significantly higher compared to the incidence of the disease in type C or D. Various etiopathogenetic hypotheses can be formulated to explain this result. Long gaps frequently characterize type A and B defects, and 100% of types A and B had been classified as long gaps in the neonatal period in the examined sample [1,2]. Long-gap defects are at greater risk of tension in the esophageal anastomosis (when feasible) and consequently at greater risk of developing anastomotic stenosis with the need for dilation and greater prevalence of GERD [22]. It is recognized that EoE and GERD are two distinct diseases, although they can influence each other, with a synergistic increase in damage to the esophageal mucosa [14]. Damage to the mucosa can be multifactorial in these patients. In the sample under examination, EoE occurs with a triple frequency in patients with long-gap defects, suggesting that the aforementioned complications caused by the difficulty of surgical correction may be involved in the etiopathogenesis of the disease. However, the isolation of the gastrointestinal system during fetal life in defects without distal fistula represents the peculiarity of type A and B defects. It could affect the prenatal development of both the mucosa and the entire esophageal, gastric, and intestinal tract wall.

More than one EA patient out of 10 was found to carry duodenal villous lesions, though just one patient received a CD diagnosis. Nine patients were therefore classified as affected by non-celiac enteropathy (NCE), none having peptic duodenitis [17,18]. NCE is reported to be a rare condition, and the exact incidence is unknown; it is calculated that NCE might cause villous atrophy in just about 5% of cases, with the remaining being attributed to CD [18]. In our population, no patient with NCE was affected by the pathological condition listed as common NCE causes. To the best of our knowledge, this high incidence of non-specific villous lesions in EA patients is firstly described. As the real incidence of non-specific villous lesions is unknown, it is hard to compare the incidence of this pathological finding among EA and the general population. Our study highlights a high incidence of microscopic changes in the duodenal mucosa. This increased prevalence of

NCE in the population with EA seems to suggest a role played by the characteristics of the anatomical defect in the etiopathogenesis of duodenal mucosal lesions. Studies have recently been published regarding the role of growth factors present in amniotic fluid during the organogenesis of the gastrointestinal tract [22,23]. As shown in Figure 1, in fetuses with EA, the defect reduces the exposure of esophageal, gastric, and intestinal mucosa to amniotic fluid and may reduce mucosal trophism during GI development [24]. Thus, it may result in a lower capacity of the GI mucosa to constitute an effective barrier.

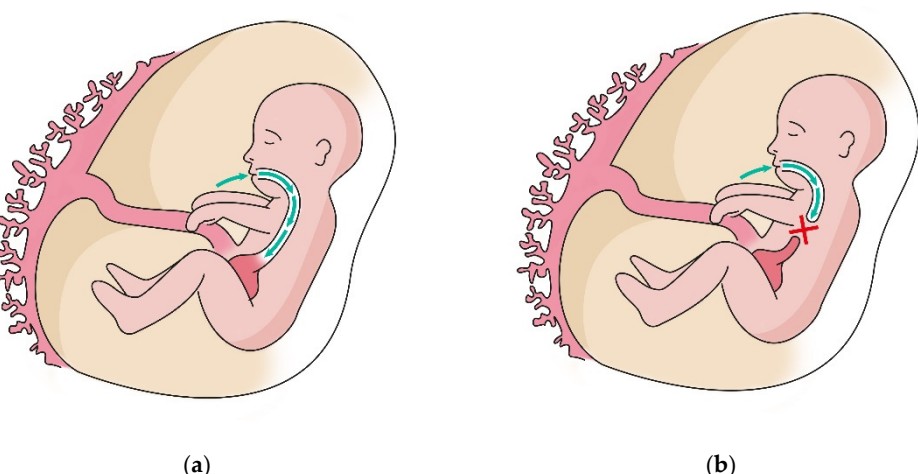

(**a**) (**b**)

**Figure 1.** Amniotic fluid circulation comprises different pathways. Fetal swallowing allows removal of amniotic fluid by mucosal absorption in the GI system. (**a**) The patency of the esophagus accomplishes this process correctly; (**b**) In case of EA, the atretic segment interrupts this process and thus remarkably reduces mucosal exposure to the amniotic fluid.

The data from our study support the hypothesis that the incidence of NCE in the EA group without distal TE fistula is approximately double compared to EA group types C and D. Although the mechanism of absorption and recirculation of amniotic fluid is known, the actual impact of the interruption of the integrity of this system and reduction or absence of contact between the mucosa and amniotic fluid remains to be fully understood.

Little is known about the possible complications faced by EA patients in adulthood [3]. Barrett's esophagus and esophageal cancer are likely to be seen in 6.4% and 1.4% of adult EA patients, respectively [25]. Our samples displayed no esophageal metaplasia or dysplasia, even if those pathological results were recorded on patients much older than ours in the literature [25]. Therefore, we consider the absence of mucosal transformation a positive finding that sustains the effectiveness of the guidelines and must be re-checked in the following years. GERD and neoplastic transformation may represent the major contributors to morbidity and mortality of EA adults [26].

Our study presents some limitations. Our retrospective study comprises results from patients already on follow-up, as well as newly acquired EA newborns. As we introduced ESPGHAN-NASPGHAN recommendations in 2016, the patients who were already in follow-up received complete histological examinations from 2016. Follow-up of every patient is ongoing, and future results will be collected and shown when available. Further studies are needed to deepen our knowledge of EA, especially long-term comorbidities.

**5. Conclusions**

EoE represents frequent comorbidity in the EA population, as previously known. Moreover, EA seems burdened by a high, never previously described incidence of non-specific mucosal lesions of the duodenum.

Due to the high incidence of clinical diseases requiring specific treatment in the EA population, embedding high gastrointestinal tract biopsies in EA endoscopic follow-up is mandatory starting from pediatric age. It must be done at scheduled timing, with

ESPGHAN-NASPGHAN 2016 recommendations being useful for clinical practice. Transitional periods must be especially monitored. Drawing a steady point of the patient's health status before entering adult life is pivotal and must be done by the pediatric specialist, who may help the adult gastroenterologist and surgeon tailoring the follow-up for every patient.

**Author Contributions:** Conceptualization, F.M. (Francesco Macchini) and F.M. (Francesca Maestri); methodology, F.M. (Francesco Macchini); validation, A.M. and M.I.; formal analysis, A.M., M.I. and F.M. (Francesca Maestri); investigation, F.M. (Francesca Maestri); resources, E.L., G.C., F.M. (Francesco Macchini) and G.F.; data curation, F.M. (Francesca Maestri); writing—original draft preparation, F.M. (Francesca Maestri); writing—review and editing, G.C., A.M., F.M. (Francesco Macchini) and M.I.; visualization, G.C.; supervision, E.L. and F.M. (Francesco Macchini); project administration, F.M. (Francesco Macchini) and E.L. All authors have read and agreed to the published version of the manuscript.

**Funding:** This study was partially funded by the Italian Ministry of Health, Current research IRCCS.

**Institutional Review Board Statement:** The present study was carried out in accordance with the principles of good clinical practice and the Helsinki Declaration, as well as the national legislative and administrative provisions in force. Due to the retrospective nature of the study, informed consent was waived.

**Informed Consent Statement:** Written informed consent from the participants' legal guardian/next of kin was not required to participate in this study in accordance with the national legislation and the institutional requirements.

**Data Availability Statement:** Data is contained within the article.

**Acknowledgments:** A special thanks to all the nurses and medical doctors of the Department of Pediatric Surgery and of the Neonatal Intensive Care Unit of Fondazione IRCCS Ca' Granda Ospedale Maggiore Policlinico.

**Conflicts of Interest:** The authors declare no conflict of interest.

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
