# Peer review of "Endoscopic Surveillance of Esophageal Atresia Population according to ESPGHAN-NASPGHAN 2016 Guidelines: Incidence of Eosinophilic Esophagitis and New Histological Findings"

_biomedicines, doi:10.3390/biomedicines10112836_

Round 1

Reviewer 1 Report

I have read with great interest this article titled " Endoscopic surveillance of esophageal atresia population according to ESPGHAN-NASPGHAN 2016 guidelines: incidence of eosinophilic esophagitis and new histological findings ", which retrospectively reviews the data of the follow-up patients operated for esophageal atresia. which retrospectively reviews the data of the follow-up of 75 patients operated on for esophageal atresia. They present data from histopathological findings collected during scheduled gastroscopies, which show an increased incidence of eosinophilic esophagitis and the occurrence of duodenal mucosal atrophy in approximately 10% of their cohort.

Here are my comments: 

•          The abstract is clear and exhaustive. 

•          At the end of the introduction the authors state “In this study, we aimed to evaluate the impact of these recommendations on the care of EA patients”. To affirm the impact, it is necessary to compare two groups, analyzing the differences before and after the introduction of the follow-up protocol. I suggest rephrasing this sentence.

•          Material and methods I was surprised as I read that the authors performed duodenal biopsies during the scheduled endoscopic follow-up. I reread the reference paper of the ESPGHAN-NASPGHAN several times (reference 3) to find out where duodenal biopsies might be suggested. I cannot find it.

•          Material and methods. Authors should specify more fully what kind of statistical analysis they performed.

•          In the results, the authors show that the incidence of EoE was higher in patients with long-gap EA. How many of these patients have received a gastric transposition? Did the authors notice a difference between patients who had received prior feeding through a gastrostomy tube?

•          The discussion is clear and covers the most significant topics, according to the most recent literature.

•          The conclusions are appropriate.

•          The bibliography is consistent and adequate.

•          The two tables embedded in the text are clear.

•          The manuscript is well-structured and comprehensible. 

In conclusion, I think that the manuscript is well structured and comprehensible, and after the suggested improvements, could meet the requirements to be published in Biomedicines.

Author Response

Dear reviewer, 

we thank you for you comments that pushed us forward to a better explanation of our work and clinical practice. 

Reviewer 2 Report

“Endoscopic surveillance of esophageal atresia population according ESPGHAN-NASPHGHAN 2016 guidelines: incidence of eosinophilic esophagitis and new histological findings”

In this article Francesca Maestri et al, performed a retrospective observational study evaluating endoscopic findings at follow-up of EA patients according to ESPGHAN-NASPHGHAN 2016 guidelines. The aim of the present study is interesting and demanded.

I have the following questions and comments to improve this paper:

  1. Materials and Methods: The patient’s history of allergy agents’ food, pollen, and fur animals. This should be clarified and added to the manuscript.

  1. Materials and Methods: According to the Updated International Consensus Diagnostic Criteria for Eosinophilic Esophagitis: Proceeding of the AGREE conference (Dellon ES, Gastroenterology, 2018) EoE is diagnosed based on a combination of 1) clinical symptoms b) histology with >15eosinophils/High Power Field c) and that other possible reasons for these symptoms and eosinophilic inflammation have been ruled out. In the present paper the authors did not rule out the presence of infections and graft-versus-host disease associated with blood hyper eosinophilia and eosinophilic inflammation in esophagus. Also, the blood eosinophil count is requested. This should be clarified and added to the manuscript.

3.      Results: In the present study 4 patients where on PPI therapy at the time for endoscopic examination. Was some of these patients on PPI treatment diagnosed with EoE? This should be clarified and added in the manuscript.

Author Response

(The authors gave the same response as above.)

Reviewer 3 Report

This work by Francesca Maestri et al. has the value of describing a large cohort of children with esophageal atresia with homogeneous follow-up and characterization with biopsies of the esophagus and duodenum.

However, I believe that a review of the manuscript is necessary, especially in terms of the concepts it handles. In particular, with regard to classifying as eosinophilic esophagitis (EoE) the high density of eosinophils in the esophageal mucosa of patients with esophageal atresia (EA), and in the affirmation that Marsh-Oberhuber grades 1 and 2 represent duodenal mucosal atrophy.

Introduction

The authors suggest a that eosinophilic esophagitis (EoE) and celiac disease (CD) based on common etiopathogenetic aspects. The literature on this topic has been confusing and full of bias in recent years, but the existence of an association between both diseases, beyond the fact that both are very prevalent and can coincide independently in the same patient, has been well ruled out by solid papers (Lucendo AJ, Arias Á, Tenias JM. Aliment Pharmacol Ther. 2014 Sep;40(5):422-34). In fact, the work that the authors use as a reference [8] already excludes the existence of an epidemiological association between both entities and consider they are fully independent. The evidence-based EoE management guidelines published in 2017 clearly stated that both EoE and celiac disease are independent entities (Lucendo AJ, et al. United European Gastroenterol J. 2017;5(3):335-358). Therefore, the authors must modify this statement in their manuscript to reflect the best current state of knowledge on the subject.

Materials and Methods

The authors assume that patients with high eosinophil counts in esophageal biopsies have a diagnosis of EoE, and base this diagnosis on the AGREE consensus recommendations. These recommendations clearly warn that EOE is a primary disease, for whose diagnosis other causes of esophageal eosinophilia should have been excluded. Regarding the particular relationship between esophageal eosinophilia and esophageal atresia, the AGREE document does not acknowledge that esophageal atresia is the cause of EoE, but literally states that this association is controversial and is beyond the scope of that article. Therefore, the authors should include a sentence saying that it is controversial to consider esophageal eosinophilia present in some cases of esophageal atresia as EoE, and name it before as “esophageal eosinophilia mimicking EoE” or even “EoE-like esophageal eosinophilia”.

On the other hand, and regarding duodenal biopsies, the authors considered a duodenal mucosal lesion classified as grade ≥ 1 of the Marsh-Oberhuber as “non-specific duodenal mucosal atrophy”. This is a conceptual error, since Marsh-Oberhuber grade I consists solely of mucosal lymphocytic infiltrate, WITHOUT villous atrophy, and in grade 2, crypt hyperplasia is added, also WITHOUT villous atrophy. Only Marsh-Oberhuber grades 3 (a, b and c) presented villous atrophy. Therefore, the unfortunate term “non-specific duodenal mucosal atrophy” should be changed to “Non-specific duodenal mucosal lesion”.

Results

Once again, a diagnosis of EoE requires that the cause of the disease be primary, and in this case this condition does not exist (eosophageal eosinophilia is like related to EA and no role for allergy has been demonstrated there). I think it would be better to call it “dense esophageal eosinophilia compatible with EoE” or “EoE-like esophageal eosinophilia”.

Once again, please correct the sentence in the results where it is stated that duodenal villous atrophy includes Marsh grades 1 and 2, because it is not true.

It would be important to also present data on HLA risk associated with celiac disease and the results of serology tests.

Discussion

Please consider changing the term EoE (which corresponds to a well-defined entity in which the EA could have a controlling relationship) to another more appropriate term

Author Response

(The authors gave the same response as above.)

Round 2

Reviewer 3 Report

In their response letter, the authors maintain that patients with esophageal eosinophilia in the context of esophageal atresia have an EoE according to the NASPGHAN/ESPGHAN guidelines. In this case, they should remove the reference to the AGREE consensus document from the top of page 3 and include the corresponding literature reference there.

Any case, and since considering that patients with atresia develop EoE remains controversial (as indicated by the AGREE statement, published in 2018 after the NASPGHAN/ESPGHAN 2016 guidelines, the discussion of the manuscript should at least mention this controversy.

Author Response

Dear reviewer, 

we thank you for your work. 

Please see the attachment for our answer. 

Kind regards
